# The Escalating Effects of Wildlife Tourism on Human–Wildlife Conflict

**DOI:** 10.3390/ani11051378

**Published:** 2021-05-12

**Authors:** Qingming Cui, Yuejia Ren, Honggang Xu

**Affiliations:** 1School of Tourism Management, South China Normal University, Guangzhou 510006, China; cuiqingming@126.com; 2South China Ecological Civilization Research Center, South China Normal University, Guangzhou 510006, China; 3School of Hotel Administration, Cornell University, Ithaca, NY 14850, USA; yr242@cornell.edu; 4School of Tourism Management, Sun Yat-sen University, Zhuhai 519082, China

**Keywords:** macaque tourism, food provision, human–wildlife conflict, community development, ecological compensation, China

## Abstract

**Simple Summary:**

Communities adjacent to protected areas usually face conflict with protected wildlife. Wildlife tourism is regarded as a tool to mitigate such conflict through bringing economic benefits to villagers and then increasing villagers’ tolerance of wildlife. We used qualitative methods to conduct a case study on a macaque tourism attraction in China and find that tourism may escalate rather than mitigate community–wildlife conflict. Provisioning food is a common way to attract wild animals to visit and stay in human activity areas. In the case of macaque tourism, anthropogenic food provision caused rapid population increase and more intra-group aggressive behaviors. More tourist–macaque interactions resulted in macaques becoming habituated to human’s presence. These ecological impacts on macaques led more invasion to the surrounding community and intensified resident–macaque conflict. Meanwhile, low community participation in tourism generated few benefits for residents and did not help alter residents’ hostile attitudes towards the macaques. Local residents gradually retreated from agriculture as the macaques became more intrusive. We propose a holistic model combining social and ecological perspectives to evaluate the role of wildlife tourism in resolving community–wildlife conflict. We suggest that wildlife tourism should minimize human–wildlife intimate interactions and food provision.

**Abstract:**

Human–wildlife conflict is a barrier to achieving sustainable biodiversity conservation and community development in protected areas. Tourism is often regarded as a tool to mitigate such conflict. However, existing studies have mainly adopted a socio-economic perspective to examine the benefits of tourism for communities, neglecting the ecological effects of tourism. This case study of macaque tourism on a peninsula in China illustrates that tourism can escalate rather than mitigate human–wildlife conflict. Fifty-three stakeholders were interviewed and secondary data were collected to understand the development of rhesus macaque (*Macaca mulatta*) tourism and community–macaque conflict. The results show that food provision and tourist–macaque interactions rapidly increased the macaques’ population, habituation, and aggressive behaviors, which led them to invade the surrounding community more often and exacerbated human–macaque conflict. Meanwhile, low community participation in tourism generated few benefits for residents and did not help alter residents’ hostile attitudes towards the macaques. Local residents gradually retreated from agriculture as the macaques became more intrusive. A holistic approach to evaluating the role of wildlife tourism in resolving community–wildlife conflict is proposed and practical suggestions for alleviating such conflict are given.

## 1. Introduction

In the past few decades, protected areas have been one of the main tools for maintaining and improving biodiversity conservation [1,2,3,4]. However, there are tensions between wildlife conservation and the development of communities adjacent to protected areas [5,6]. The establishment of protected areas deprives communities of natural resources and restricts industrial and agricultural development, suggesting that to conserve ecology and wildlife, those communities sacrifice economic opportunities [7,8,9]. Moreover, wild animals often cross the borders of protected areas and enter neighboring communities, causing human–wildlife conflict [10,11]. The costs that wildlife impose upon local people include crop-raiding, livestock loss, human attacks, and opportunity and transaction costs [8,12]. Local residents who suffer economic, social, and health losses may then become hostile to wildlife and conservation, and even harm or kill wild animals for revenge [12,13]. Human–wildlife conflict is therefore one of the main problems besetting sustainable wildlife conservation and the sustainable livelihoods of local communities. “Human–wildlife conflict” in this study mainly refers to the community–wildlife conflict, following most other conservation studies, e.g., [12,14,15].

Wildlife tourism development has been proposed as a solution to human–wildlife conflict [3,14,15]. Recent studies have focused on examining whether and how tourism benefits can alter communities’ hostile attitudes and livelihoods from the economic and social perspectives [16]. In this article, we argue that those studies neglect the ecological costs of wildlife tourism. Human–wildlife interactions in tourism can bring about various adaptive ecological and behavioral changes that cause wildlife to become a nuisance and make human–wildlife conflict difficult to manage [17,18]. The introduction of profit-driven wildlife tourism in protected areas can trigger complicity in relation to human–wildlife conflict and result in a divergence from original conservation principles. In order to bridge the above research gap, we propose a holistic approach that synthesizes social and ecological perspectives to examine the interactions among tourism businesses, local community, and wildlife.

We use macaque tourism in Hainan Province, China, as a case study to show how wildlife tourism can intensify, rather than mitigate, human–wildlife conflict. The specific research questions include: (1) How do the community residents cope with the community–wildlife conflict? (2) How does the community participate in tourism, and can tourism benefits change the community’s attitude towards wildlife? (3) How does tourism activities affect wildlife? (4) Does wildlife tourism exacerbate or mitigate human–wildlife conflict if assessing the socio-economic benefits and ecological costs combined?

## 2. Literature Review

### 2.1. Tourism as a Way to Mitigate Human–Wildlife Conflict

There are controversial arguments about whether and how tourism development mitigates human–wildlife conflict in protected areas. Many studies endorse the premise that tourism benefits that accrue to local residents can raise villagers’ environmental awareness, increase residents’ tolerance of wildlife [15], and transform traditional livelihoods [19]. For instance, Mbaiwa and Stronza found that in Okavango Delta, three communities participating in tourism had stopped traditional activities, such as hunting, gathering, livestock raising, and crop farming [20]. Tourism revenue-sharing projects in gorilla tourism in Rwanda and Uganda have received the most research attention, with studies finding that national park officials and local representatives believe that revenue-sharing is the most significant advantage of living adjacent to gorilla national parks [21] and that residents benefit from tourism revenue-sharing through infrastructure projects [22,23]. Cases from Brazil and Peru also support the argument that tourism benefits local residents, and that local participation in management can generate conservation attitudes and actions [24].

However, other scholars have questioned the effectiveness of tourism in improving community development and biodiversity conservation [9,16,25]. Swemmer et al. pointed out that “benefit sharing is messy, is complex, and occurs at various scales with multiple trade-offs” [26] (p. 17). Stakeholders at different scales have heterogeneous demands for revenue, for various reasons. For instance, communities in Uganda’s Mgahinga Gorilla National Park were found to want benefits to compensate for crop and livestock losses caused by wildlife, park officials hoped to use tourism revenues to offset the costs of management, and the national government tended to allocate tourism revenues according to the conservation needs of the whole state [27]. In many developing countries, the government and park authorities have the power to determine the allocation of tourism revenues, and communities lack access to participation in the decision-making process [25,28,29].

In addition to unequal power relations and a lack of local participation, the uneven distribution of tourism benefits is another problem [16]. Within communities, poor residents often perceive that the elite obtain the majority of tourism benefits [22,28]. Hemson et al. noted that only residents in tourism industry gain benefits; most local residents are not beneficiaries of tourism [30]. Tourism revenues may not be distributed evenly among different communities. For example, residents in the buffer zone of Nepal’s Chitwan National Park are constrained in their use of natural resources, but only those villages close to the park’s entry points benefit from tourism incentives [31]. In China’s Wolong Giant Panda Nature Reserve, the communities close to main roads gain more income from tourism than the remote communities that bear greater costs of conservation [32]. The spatial unevenness of revenue distribution has also been found in gorilla national parks in central Africa [25,33,34].

Moreover, the distribution of tourism economic incentives is often mismatched. Crop-raiding and livestock loss caused by wildlife are the problems of most concern to local residents [25,34]. However, tourism benefits are often allocated to improving social infrastructures, such as clinics, schools, roads, bridges, wells, and water tanks, rather than direct compensation or the prevention of human–wildlife conflict [21,22]. Because of this mismatch, local people remain hostile towards wildlife [30]. The locals in Kibale National Park in Uganda regard building elephant trenches as being better than building schools and roads [35]. Many scholars have therefore suggested that tourism revenues should be used to directly offset losses caused by human–wildlife conflict to more effectively improve local attitudes to conservation [22,25,28,34].

Generally, existing studies mainly evaluate the role of tourism in conservation from the socio-economic dimension, and conclude that although tourism benefits contribute to changing local people’s attitudes towards wildlife and conservation, the effectiveness of tourism is limited due to the unequal, uneven, and mismatched distribution of benefits [16,36]. By contrast, few studies consider the ecological impacts when assessing the impact of tourism on mitigating human–wildlife conflict. Many protected areas use their unique species as tourist attractions to generate economic revenues. However, wildlife-based tourism development is not without cost. Tourism activities can generate negative effects on wild animals [37,38], which should also be considered when evaluating social and economic benefits [39]. In the next section, we review the effects of wildlife tourism on macaques as an exemplar species.

### 2.2. The Effects of Wildlife Tourism on Macaques

Humanity has a long history of interacting with macaques (*genus Macaca*). In contemporary society, free-ranging macaques have become popular tourist attractions. There are 23 species of macaque distributed in Asia, North Africa, and Gibraltar, many of which are strongly involved in the tourism industry [40].

Wild animals usually avoid encountering humans, which makes wildlife-based tourism unpredictable and uncontrollable [41]. In non-captive macaque tourism, food provisioning is a common way to tempt macaques to stay in a certain area and become habituated to the presence of people [42,43,44]. In some wildlife tourism sites, feeding animals is itself an important tourist experience [44]. Provision of food is an effective strategy to increase the likelihood of tourists interacting with free-ranging macaques [44].

However, food provisioning and tourist activities have various negative effects on macaques [44,45]. Anthropogenic foods are highly caloric and are more palatable and more accessible than macaques’ natural foods. Wild macaques therefore spend more time at tourist sites and come to rely on the provided food supply, resulting in changes in their activity budgets and dietary diversity [43,46]. Provisioning also affects the population in diverse ways. A stable, intensive, and abundant food supply can dramatically increase the population of macaques [39,44,47]; however, close contact raises the possibility of mutual pathogen transmission between humans and macaques, which can further affect the health and population of the animals [48].

Macaques in tourism areas gradually develop interspecific aggressive behaviors. The presence and proximity of tourists can elevate Barbary macaques’ anxiety levels [49,50]. Many tourists are not satisfied with inactive wildlife. They often tease monkeys to behave more actively by pointing, waving, slapping, mimicking, yelling, throwing food, and even threatening [50,51,52,53]. Many studies show that tourists initiate the majority of interactions with macaques [54,55]. Tourists’ provocative behaviors induce monkeys’ agonistic responses such as biting, scratching, hitting, and threatening [52,56]. Because of interspecific differences, tourists generally misinterpret the meaning of monkeys’ behaviors, which may enhance the agonism. For example, Maréchal et al. found that in interactions, most tourists cannot identify the exact meanings of macaques’ facial expressions [57]. The longer the history of visitors’ interactions with macaques, the more aggressive the macaques may become [58].

Food provision also intensifies intraspecific agonism. Monkeys fight with each other during the feeding time [59]. There is a positive correlation between food provision and the frequency of in-group aggression [54]. Furthermore, tourists like to feed baby monkeys, which they perceive as cuter than adults [51]. This preference violates the strict hierarchy among macaque groups and increases the rate of attacks on baby and juvenile macaques by male adult macaques [51,59].

Despite these negative impacts, some scholars regard human–macaque interactions in tourism as opportunities that have stimulated the evolution of macaques [18]. Evidence for this point of view is the robbing and bartering behavior developed by long-tailed macaques at Uluwatu Temple, Indonesia [60]. The macaques have learned to steal inedible objects such as glasses and hats from tourists and barter the objects for food with the staff. This innovation has been socially learned and has spread in the group, suggesting that human–macaque interactions in tourism can cause significant cultural change in a macaque group [61].

Existing research shows that tourism affects macaques at the population, behavioral, and cultural levels. Macaques can develop adaptive behaviors in anthropogenic tourism environments. Barrett, Stanton and Benson-Amram called for more studies to explore the roles of animals’ adaptive behaviors in worsening or mitigating human–wildlife conflict in protected areas [17]. This study uses macaque tourism in China to show that the effects of tourism on macaques can exacerbate rather than mitigate human–wildlife conflict.

## 3. Materials and Methods

### 3.1. Study Site

The study site is Nanwan peninsula in Lingshui county, Hainan, China (Figure 1). This peninsula consists of three main areas: Nanwan Macaque Provincial Nature Reserve, Monkey Tourism Park, and Nanwan Village. The nature reserve was established in 1965 and covers 10.2 square kilometers [56]. The reserve contains more than 2000 rhesus macaques (*Macaca mulatta*), which are second class protected animals in China. There is a protection station responsible for conservation work. In 1974, a tourism park was built in the experimental zone of the nature reserve. Food was used to attract wild monkeys into the tourism area [62]. In 2020, more than 500 monkeys visit the tourism park every day, and approximately one million tourists visit every year. Nanwan Village, which has approximately 550 residents, is also located in the experimental zone of the reserve. Agriculture is still the way of life for some villagers. Macaques often cross the border of the nature reserve and enter the village, causing community–macaque conflict.

The macaques are the only target attraction in this area. There are also some potentially attractive houseboats on the sea, where some water people still live. However, those houseboats are usually seen from the cable cars, few tourists approach them.

### 3.2. Data Collection

The research team visited the site 10 years ago and conducted a study attempting to understand the tourism development model of the conservation area. The current qualitative study is based on twice fieldwork conducted from 16 to 22 February 2019, and from 28 to 30 September 2020. Data were collected using semi-structured interviews and observation. We used the method of purposive sampling to find the people who best know the situations about tourism, community, and nature reserve. We interviewed 2 managers from the nature reserve, 2 managers and 5 staff from the Monkey Tourism Park, 26 tourists, the chairman of Nanwan Village and other 17 Nanwan villagers. We also interviewed a biologist who had studied the macaques in the park since 2013. Interviews with tourists were mainly conducted at the visitor center. The main questions were about the visitors’ general views on macaques, and how they perceived and reacted to aggressive macaques. Interactions between the tourists and macaques were observed and recorded as field notes. Interviews with managers and staff were conducted in their workplaces. We mainly asked about the development of scenic spots, management of the macaques, and community participation in tourism. The interviews with the nature reserve managers covered the establishment and development of the protected area, the protection of macaques, the relationship between the protected area and the tourism park, and responses to community–macaque conflict. Interviews with the villagers concerned their livelihood, their attitudes to macaques, and management of the nature reserve and tourism park. All interviewees gave their permission to be recorded.

Observation was mainly used to understand the spatial arrangements of tourism park, tourist routes, nature reserve, and community land utilization. For example, the route that tourists go to the park from the mainland and return, the distance between the community and the park, the locations of village mango groves. The spatial relations between the tourism park, nature reserve, and village are essential to understand human–macaque conflict (see Appendix A).

In addition to the above first-hand data, we also searched second-hand data about the Monkey Tourism Park, such as published research articles (e.g., [56,62,63,64,65]), and news reports (e.g., [66,67]) to help comprehensively understand the history of tourism development and macaque protection.

### 3.3. Data Analysis

From numerous qualitative data analysis methods, we chose “thematic analysis” [68] to analyze our collected materials. During the data analysis, the audio records were transcribed first. Then, two authors separately read the first-hand and second-hand data repeatedly to get familiar with the data. Second, we generated many initial codes about conservation conditions, community-macaque conflicts, community participations in tourism, and tourism’s ecological impacts. Third, we thought about the relationships between codes and categorized these codes into many sub-themes and themes, including conservation modes, villagers’ strategies to cope with conflict with macaques, low community participation in tourism, and two main ecological impacts of tourism on macaques. Fourth, each of the two authors reviewed and named the themes. After that, we wrote an outline by relating these themes to explain the story of community–macaque conflict and tourism development, then compared the two outlines to obtain a mutually agreed version and construct a thematic map. The third author then compared this outline with the data to check its validity, and proposed a final thematic map (see Appendix A), on which the results are based. Finally, the three authors proposed a general model to explain the exacerbating effects of tourism on human–wildlife conflict according to the evidence from Nanwan.

## 4. Results

### 4.1. Coercive Fortress Conservation and Spatial Exclusion of the Community

The community–macaque conflict on the Nanwan peninsula has existed for a long time. In the 1930s and 1950s, before the establishment of the nature reserve, the conflict was solved at the cost of a loss of macaques. To safeguard their crops, community residents killed macaques. When the nature reserve was established in 1965, there were only 5 groups of macaques left, comprising about 115 individuals [62].

The nature reserve system in China prohibits any use of natural resources in the core and buffer zones, and only allows limited research, education, tourism, and leisure activities in the experimental zone [69]. This management regulation tallies with the model of fortress conservation, according to which “biodiversity protection is best achieved by creating protected areas where ecosystems can function in isolation from human disturbance” and “only tourism, safari hunting, and scientific research are considered as appropriate uses within protected areas” [70] (p. 704). The fortress conservation in Nanwan is coercive and underpinned by national laws. All conservation work in the reserve is run from a protection station, which routinely sends rangers to patrol and record at various points in the reserve. Considering that the Nanwan villagers and their ancestors have lived in this area for a long time, this conservation model excludes the community residents from using resources that once belonged to them. It also means that Nanwan villagers sacrifice their development opportunities for conservation.

As a result of the coercive fortress conservation, macaques are well protected. The number of macaques has undergone a rapid increase. In 1988, there were 903 macaques in Nanwan nature reserve [63]. In 1998, the population was estimated to be 1300 [63]. In 2019, the manager of the nature reserve told us that the current estimate is more than 2000 macaques.

### 4.2. Community–Macaque Conflict and the Lack of Ecological Compensation

As the population of macaques has grown, community–macaque conflict has worsened. According to Lian and Jiang [64], the ecological capacity of Nanwan nature reserve can provide resources for 1900 macaques at most. The current macaque population level has exceeded the maximum capacity. In a study conducted in 2010 [65], an ecologist has pointed out the problem of ecological overshoot on Nanwan peninsula.

Many bold macaques now enter the community area to search for food. The most damaged crops include mangoes, sweet potatoes, and watermelons. In Nanwan village, nearly every household used to have a mango grove, and selling mangoes was one of their main income sources. When the mango harvest was better, a farmer could earn about $2800 USD to $4200 USD per year. However, when the mangoes are ripe, macaques enter into the groves almost every day. As one resident (L02) described: “We are the poorest village in this town area. When mangoes ripen, macaques come down to eat. They not only eat whatever they can, but also grab and throw away the rest.” Some monkeys have even broken into residents’ houses to search for cooked food or steal eggs from chicken pens. The locals show obvious hostility towards the monkeys by describing them as “public nuisances” and “thieves”.

Because of the legally protected status of the macaques, the community cannot hurt monkeys as their ancestors did in the past. After the establishment of the nature reserve, it was made very clear to the Nanwan villagers that capturing monkeys is illegal. Nowadays, the residents do not have effective ways to expel the annoying macaques. “Many macaques come to the village at a time. You cannot catch them. You cannot beat them. We know it is illegal. If we frighten them, they run to the top of the trees and cannot be driven away. They are animals, we cannot control them.” (L01). Some villagers tried to isolate mango groves from the macaques using nets, but staff from the protection station stopped that defense because they feared that the net may pose a threat to the macaques. The conflict between the community and the macaques became more tense.

As a result, villagers were eager to be compensated by the government for their loss of livelihood. However, the protection station manager said: “There is no special fund for ecological compensation in Hainan Province for macaque damage.” (SM01). Nanwan villagers complained about the lack of ecological compensation: “We live on mangoes, macaques often come down to eat, they [the protection station] do not give us money, even a penny.” (L02). With no ecological compensation, the conflict between villagers and macaques remains unsolved, even though the station manager is aware of the macaques’ crop-raiding.

### 4.3. Monkey Tourism and Limited Community Participation

The development of monkey tourism brought a new potential opportunity to solve the conflict between villagers and macaques. There is a long history of developing macaque tourism on Nanwan peninsula. In 1974, the staff of the protection station began to feed two groups of macaques and the area received tourists in 1980 [62]. Around 1985, the Lingshui county government established a tourism company and cooperated with the reserve to develop monkey tourism on a large scale [62,66]. After experiencing 10 years of rapid development, the tourism park began to decline [67]. To restore tourism development, in 1999 the county government sold the rights of developing the park to a private cable car company. To reinvigorate tourism development, the company built a two-kilometer cableway to connect the mainland to the peninsula, rebuilt the park’s infrastructure, and improved the park’s management [67]. The tourism development has followed the conservation plan, such that nobody can enter the core zone of the reserve; the tourism park is restricted to the experimental zone and tourists can only interact with macaques when the monkeys freely enter this zone [65]. The attraction is now a national 4A level scenic area and attracts approximately one million tourists every year.

However, the prosperity associated with the monkey tourism has not resulted in the economic development of Nanwan village for many reasons. The most important is that the enclaved mass tourism model leaves no room for business opportunities for the Nanwan villagers. Most visitors are package tourists. They enter the scenic area via the sightseeing cableway from Xincun town on the mainland. The trip is about two-hour visit. They then either take a cable car or a boat and shuttle bus back. There are no tourism products in the village. Thus, there are few intersections between the tourists’ activities and village spaces. The villagers once applied to do business in the scenic area, but were denied by the manager. “When the scenic area belonged to the county government, we could sell things inside. When the tourism company contracted with the local government, they promised we could still do business inside. But after selling for one month, we were driven away by the tourism company and were not allowed to sell things from then on.” (L08).

Second, the tourism park only employs a limited number of villagers, due to its adoption of a modern business management model. Well paid and skilled jobs can only be from outside the peninsula. There are more than 130 households in Nanwan village. Only about 10 of them have obtained low-salary jobs in tourism to do cleaning and security work. The average salary is about $230 USD per month, lower than working in other places. “Now only some old villagers work in the scenic spot as security guards. They look after the scenic spot during the night.” (L09). Work in tourism is not economically attractive compared with off-farm jobs in the city.

Third, about 10 households rent their land to the company managing the scenic area. However, the contract was signed 20 years ago for a period of 50 years with a land value based on its 1999 evaluation. The village chairman revealed that: “About 10 households have contracts with the scenic area. Rents are low. Now the park pays the rent every year, about $50 USD per household. Only about 10 households receive the money.” (L01). The residents are powerless compared to the company, who is a big tax-payer for Lingshui County. Thus, it is impossible for the residents to push the company to sign a new contract based on the current land value.

The perception of the tourism park manager is that the company is not responsible for community development and compensation for macaque damage. “I am not clear about the negotiation between the nature reserve and the community. It [the compensation] has nothing to do with our park. We are only responsible for business operations. Macaques belong to the nature reserve, who should be responsible for the compensation.” (PM01).

It may be true that for historical reasons, the partnership among the conservation committee, the park company, the village committee, and the villagers was not perfectly designed because the potential dynamics of tourism and the macaques were not clear initially. The major challenge now is that the initial collaboration model was not designed to enable all parties to negotiate and benefit from the collaboration when changes occur. In the next section, we show how the development of macaque tourism is likely to escalate community–macaque conflict, and demonstrate the obligation of the tourism industry to provide compensation.

### 4.4. Adaptive Macaques and Escalation of Human–Wildlife Conflict

#### 4.4.1. Population, Aggression, and Tourist–Macaque Conflict

As with macaque tourism in other regions, developing tourism on Nanwan peninsula has had many ecological effects on monkeys. One of the most obvious impacts is the rapid growth of the macaque population. Food provision is the main method of attracting wild macaques to the tourism park area and keeping them there. Such provisioning started in 1974 and continues now. Usually, the macaques come down from nearby hills every morning, stay in the park area during the day and go back to the hills in the evening. The park staff feed the macaques at 08:00, 12:00, and 17:30 each day with wheat and various vegetables [56]. In 2013, around 80 g of food was formally provided per individual per day [71]. In addition, the park sells food for tourists to feed the macaques. Anthropogenic provision can increase macaque populations. A biologist who is studying macaques in the park said that “provision can lead to the growth of the monkey population within the park. My records are from 2013 to this year 2019 and show that the number of fertile adult female macaques has grown every year in this park.” (R1). More fertile adult female macaques mean that there are more baby macaques every year. Zhang et al. recorded 7 groups with approximately 350 individuals visiting the park in 2014 [56]. According to the official data, the macaque population grew from 393 in 2018 to 433 in 2019. The head of Department of Macaque Management estimated that the number is more than 500 in 2020 and there are about 80 new-born baby macaques every year. The population growth effect of tourism provision corresponds with studies of Barbary macaques in Gibraltar [47] and Japanese macaques in Oita [39].

Second, the macaques become habituated to the presence of people and more aggressive to tourists. The provisioned food (80 g/individual/day) is not enough to satisfy the needs of every monkey. Many monkeys develop robbing, biting, scratching, and threatening behaviors towards tourists. Zhang et al. recorded 195 instances of aggressive behavior, most of which were aimed at obtaining food [56]. Tourist-induced aggressive behaviors accounted for 54.67% of the total [56]. Visitors often pursue close interactions with monkeys by feeding and touching them, even though the park regulations and tour guides prohibit such behaviors.

During our field observations, we found that when entering the park, tour guides usually counselled visitors not to feed monkeys or hold drinks, foods or colorful bags in their hands, and they repeatedly reminded visitors not to open bags in front of the macaques to prevent robberies by the monkeys. Park managers also set up many noticeboards to remind the tourists not to engage in these “transgressive” behaviors, but still allowed tourists to buy park-provided food to feed the macaques. Tourists usually perceived the monkeys, especially baby monkeys, as “funny, cute, and human-like”, and thus “approachable and playful” (T01). Many tourists ignored or forgot the guides and warnings, and approached the macaques to feed, tease and interact with them. These behaviors are highly likely to incur an attack. Staff in the park’s clinic said that nearly every day there are incidents in which tourists are hurt by macaques. Moreover, there were 45.33% aggressive behaviors initiated by the macaques [56]. Sometimes monkeys actively rob food, drinks, and inedible objects such as paper napkins and glasses from careless tourists. Some bold adult male macaques may even open tourists’ bags to search for food.

The above evidence shows that macaques in Nanwan Monkey Park have developed adaptive aggressive behaviors through long-term human–macaque interactions to better obtain anthropogenic resources and maximize their benefits. Tourists who are bitten or scratched by macaques need to be given vaccines, which increases the economic cost of the park’s operation and leads to economic disputes with tourists and the tourism company. Therefore, the park has built many wooden “cages” for tourists to eat inside to avoid monkey robberies. A number of security guards have been recruited to constantly remind visitors to pay attention to safety, and stop tourists from engaging in transgressive behaviors. The security guards are also responsible for driving macaques back to the trees when they gather at the visitor center or trails. However, it is difficult for the guards to watch over such a high number of tourists and troublesome macaques. The interviewed biologist revealed that in recent years the park had decided to increase the amount of food provision with the aim of making the macaques fuller and thus reducing their attacks on tourists. Now, in 2020 the head of Department of Macaque Management told that the park fed 50 kg of rice and 10 kg of peanuts for the macaques every day, that is 100 g for an individual per day.

Many measures have been formulated to mitigate tourist–macaque conflict in the park. However, the impact of the aggressive monkeys on the local community on Nanwan peninsula has been largely neglected. The monkeys not only make trouble in the park, but also invade Nanwan village, intensifying community–macaque conflict.

#### 4.4.2. Intensification of Community–Macaque Conflict

It is difficult for the rapidly increasing and aggressive macaques in the tourism park not to influence the community, because the park is next to the community’s mango groves and the macaques are free to range in the park, the nature reserve, and the village. The park is separated from one main piece of village’s mango groves by a wall, which “can only prevent humans entering the park, not stop the monkeys entering the mango grove”, commented by a villager (L16) (see Appendix A). The interviewed biologist revealed that “Nanwan village is just next to the monkey park. Sometimes park macaques do range in Nanwan Village.” (R01). Although there has been no quantitative research calculating the proportions of park macaques and nature reserve macaques invading Nanwan village, it can be sure that monkey tourism has affected the community–macaque relationship.

The effects of tourism on the population and behaviors of the macaques have contributed to the escalation of community–macaque conflict. First, the fact that the tourism park has provided a stable and increasing food source for macaques has made the park a “reservoir” of constantly reproducing macaques. These macaques inevitably overflow into the surrounding areas including the village, especially when the park does not provide abundant food. A resident (L07) commented that “A few years ago there were more monkeys because they [the park] did not provide enough food for the monkeys, who then came to the village to steal food. Now, they feed more food and monkeys have become fewer in our village.” The village head told that “There were more monkeys disturbing the village in the COVID-19 pandemic compared to past years, maybe because the tourism food provision became less” (L01). The problem is that more provision can only keep macaques in the park temporarily, at the cost of macaques reproducing in the future. Thus, it is predictable that more provision cannot resolve the conflict between the community and the macaques, but will eventually intensify it. Second, the park macaques have become habituated to the presence of humans. When they invade Nanwan village, they are not afraid of villagers. Hence, villagers have few methods to repel the macaques. “After our protection, the macaques live harmoniously with humans. When people treat macaques well and stop hurting them, their courage will increase and they will definitely come to the village to find food to eat.” (SM01). The ecological impacts of tourism on macaques complicate the community–macaque conflict.

Residents’ reactions to macaque invasion provides further evidence of the escalated conflict. Nanwan villagers usually adopt two strategies in response to the ongoing macaque invasions. Some villagers spend more time and energy watching out for macaques to defend their mangoes (see Appendix A). In the ripe season, they have to stop most other daily work to drive away the invading monkeys, who may appear in the village at any time. Although driving away the monkeys is quite time and labour consuming, this intense defense does not significantly reduce invasion by the macaques. For instance, “Near the harbour, a boss rents the land and has planted many mangoes. The operation is different. They employ special men to drive away macaques all the day. But they said there are still a lot of monkeys going to eat mangoes.” (L06). Even the professionalization of repelling monkeys cannot stop macaque damage. Thus, many other residents have given up planting in most of the mango groves near the hills and the park (see Appendix A). Some have even abandoned mango cultivation as a main livelihood and have chosen to find jobs in the county town. “Now my family members don’t care about whether the mangoes grow well or not. We find jobs in other places. My parents stay at home, but have also stopped planting mangoes.” (L04). Only about 15 households still plant mangoes now, according to the estimate of the village head. The strategies of intense defense against macaques and retreating from planting mangoes are the community’s helpless reactions to the escalation of community–macaque conflict.

## 5. Discussion

The Nanwan case shows a dynamic and growing pattern of human–macaque conflict over time. This conflict existed when the community lived on the island before the reserve was established and did not stop when the nature conservation system was implemented. However, conflicts intensified when tourism was introduced. The structures leading to these conflicts are presented in Figure 2. We argue that making food available to macaques is the critical aspect underlying all of these complicated conflictual relationships.

In the absence of anthropogenic and external influences, wild macaques live mainly by foraging for natural food. Under this condition, the macaque population will not exceed the maximum ecological capacity, and in the long term, wild macaques and the natural environment will reach an ecological balance. As the top left loop in Figure 2 illustrates, natural food resources impose a constraint on the growth of macaque populations, and vice versa. In the loop without anthropogenic influence, *ecological rule* plays the vital role in controlling macaque population.

However, in reality there are usually human communities adjacent to protected areas, and their agricultural crops provide additional food sources for wildlife. Hence, when protected areas lack sufficient food, wild macaques will invade the surrounding communities for extra food. When crop raiding, macaques must bypass communities’ guards, which usually evolves into human–macaque conflict. In the case of Nanwan, the villagers have developed different strategies to cope with human–macaque conflict in different social, historical, and institutional contexts.

Before the establishment of the nature reserve and the legislation of macaque protection, humans had advantages over the monkeys. They often hunted the transgressive macaques to protect their crops, which reduced the wild macaque population. After the nature reserve was established, the advantages were reversed. The villagers were prohibited from hurting the macaques, even for the purpose of protecting their property. Subsequently, the residents adopted a defensive strategy in the short term and attempted to drive the nuisance macaques away. This defense was effective in reducing crop loss and restraining the rapid increase of macaques. However, because of the gradual habituation of the macaques to the presence of humans, the residents have become unable to find effective methods to expel the monkeys once and for all. Thus, their defense strategy has failed to stop the macaque invasion. Human–macaque conflict has heightened; crop damage has increased, and the macaque population has grown steadily. In the long term, as the labour and time costs of defense have continued to increase, the community has begun to retreat, stop planting crops and find alternative livelihoods, resulting in a further increase of the macaque population in the community’s agricultural area. The *conservation policies and community’s livelihood strategies* determine the community–macaque interactions. In the context of coercive fortress protection, macaques have big advantages over humans, and the community–macaque conflict loop will not be completely mitigated until the community fully retreats from agriculture.

The introduction of wildlife tourism was supposed to mitigate community–macaque conflict. However, food provision has exacerbated the conflict. The Monkey Tourism Park uses food to tempt wild macaques to visit the park regularly, as many other tourism attractions have done [39,51,52]. This attracts tourists who want to watch and interact with the monkeys at close range. Close tourist–macaque interactions induce frequent macaque attacks on tourists [52,56], which have resulted in additional economic costs and disputes for the tourism park. To reduce these attacks and their associated costs, the tourism park has increased the amount of food provided to make the macaques more satiated and thereby stop them robbing food from tourists. Thus, a vicious loop has developed. *Commercial logic* dominates this loop. For maximizing the profit, the tourism park must maintain and increase food provision to attract macaques coming and reduce macaque attacks. As a consequence, the population of macaques will keep growing [39,47]. Macaques may also develop adaptive behaviors, including aggressive behavior, in continuous interactions with humans [18,61].

As the macaque population grows, the food provided by the tourism park can never be enough. Some macaques inevitably intrude into the nearby village to search for edible crops. This widens the macaque–community conflict loop. Because of the constantly reinforced loop in the tourism park, crop damage is intensified as more and more macaques become habituated to and aggressive towards humans. As a result, the macaque population gradually increases and the community retreats from agriculture with a hostile attitude.

This integrated model provides a general theoretical explanation of the dynamic and growing pattern of human–macaque conflict under the impact of wildlife tourism. The model covers the interactions between tourism, the community, and the macaques that we can see. However, it needs more testing and validation before it can be extended to explain conflict between humans and other species in other regions.

## 6. Conclusions

In protected areas, human–wildlife conflict is one of the main barriers to achieving sustainable biodiversity conservation and community development [6,12]. Tourism is usually regarded as a way to mitigate human–wildlife conflict by involving communities in tourism to obtain benefits that will change their hostile attitude towards wildlife or transform traditional livelihoods [3,15,20]. Existing studies mainly focus on examining the effectiveness of tourism in mitigating human–wildlife conflict from social, economic, and political perspectives [16,36], ignoring the biological consequences to wildlife. This study contributes to understanding the effect of using tourism as a tool to mitigate human–wildlife conflict. The provision of food in wildlife tourism can dramatically change the natural ecosystem and the behaviors of macaques, as well as the balance between natural capacity and the number of macaques. Human–wildlife conflict worsens in this scenario.

This research suggests the need for a holistic, integrated, and dynamic approach to evaluating tourism development and solving human–wildlife conflict in protected areas. Human–wildlife conflict is not only embedded in social systems, but also involves ecosystems. Thus, it is not adequate to solely consider the social and economic benefits brought by tourism. It is also necessary to consider tourism’s impacts on wildlife and ecology. As wildlife habitats shrink, more wild animals will cross the borders of protected areas into peripheral spaces, which then become the hot spots at the human–wildlife interface. It should be recognized that it is nearly impossible to protect a species without any human disturbance in the Anthropocene [72]. Hence, a balanced approach is to assess the mitigating effects of wildlife tourism in human–wildlife conflict by integrating social and ecological/biological perspectives.

Moreover, scholars should pay more attention to the agency of animals. Animals are not passive things waiting for human actions. Wildlife can actively develop adaptive behaviors in anthropogenic environments to maximize its own benefits [18]. More human–wildlife interactions provide more opportunities for wildlife to acquire more adaptations [17], such as the robbing and bartering behaviors of long-tailed macaques in Bali [61]. This adaptation and evolution, when combined with coercive institutional protection, give wildlife many advantages in conflicts with humans and make the conflict unmanageable and uncontrollable, as in the case of Nanwan. Wildlife tourism is based on human–wildlife interactions, and their impact on the adaptation and evolution of wildlife should be fully considered.

### Implications

The integrated model of tourism–macaque–community interactions proposed in this study needs validation for other species and in other regions. Trans-species interactions in tourism can generate different effects on different wildlife due to the heterogeneity of species [37]. Thus, wildlife may evolve heterogeneous behaviors and then influence community–wildlife conflict in different ways. For instance, human–elephant conflict is also a challenging problem due to the excellent memory ability of elephants and their high food demand. Moreover, the diversity of social, cultural, economic, and political contexts in which human–wildlife conflict is embedded means that different regions must adopt different strategies to cope with the conflict. Under China’s current policies, culling protected wildlife is not possible to control wildlife populations, but in some other countries the culling of overpopulated wildlife is a normal ecological management policy. Different social, cultural, and institutional contexts produce different ways to address human–wildlife conflict.

This study has many implications for practice. Our observations suggest that wildlife tourism attractions should design their activities cautiously and minimize human–wildlife interactions and food provision if possible. It is better to control tourists’ behaviors to meet the behavioral patterns of wildlife rather than the other way around. If the tourism sites plan to feed wildlife, it is better to provide food via formal supervised arrangement and eradicate tourists’ accidental and non-regulatory feeding activity [44]. Furthermore, the institutional design for cooperation among communities, tourism companies, and conservation committees should also be flexible to allow for the adjustment and re-negotiation of benefits and obligations, because the interaction among tourists, wildlife, community, and ecosystem is always evolving. Because, in general, local communities are in relatively weak bargaining positions, it is better for governments to establish special ecological compensation projects for wildlife damage, and control wildlife populations properly according to the principle of maintaining ecological balance.

## Figures and Tables

**Figure 1 animals-11-01378-f001:**
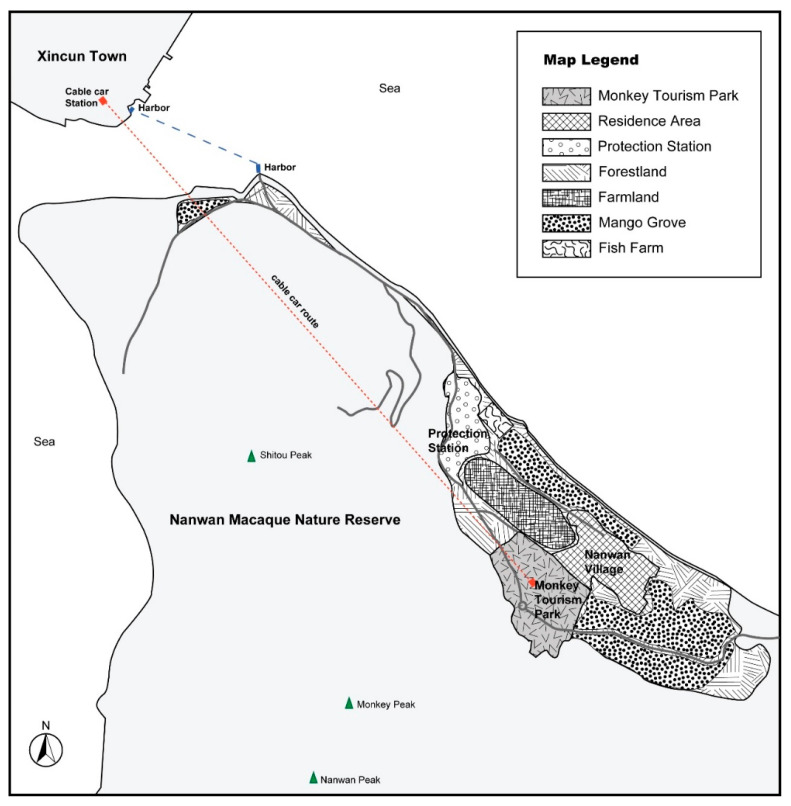
Map of the study site.

**Figure 2 animals-11-01378-f002:**
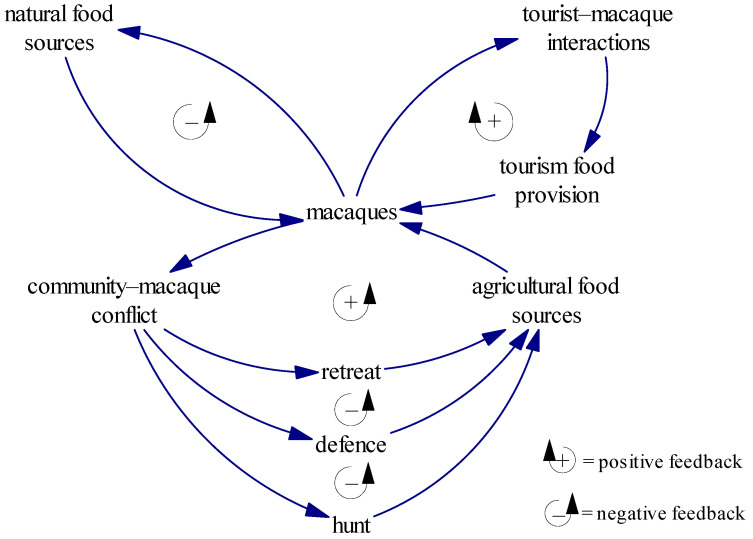
Integrated model of tourism–macaque–community interactions.

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
