# Peer review of "The Escalating Effects of Wildlife Tourism on Human–Wildlife Conflict"

_animals, 2021, doi:10.3390/ani11051378_

Round 1
Reviewer 1 Report
The authors present an interesting study where wildlife tourism has escalated conflict with a local community rather than de-escalating it through an economic or other benefit. Tourism is directed at a single species, a macaque. Given that they authors state (line 137) that there are 23 species of macaques it would be useful if the study species was named. Furthermore, it would be useful to know if the site has other attractions that might disperse the tourists and contribute to benefits or costs. The key features of the study are (1) the macaques receive supplementary feeding sustaining populations that may be above the site’s carrying capacity (the ecological effect), (2) the site managers have responded to threats to visitors and the neighbouring community by increasing the supplement which is driving a positive feedback to (1) with further adverse consequences, and (3) the neighbouring community has not been compensated for macaque incursions and have in fact been dispossessed from economic benefit by other commercial interests. The topic of feeding wildlife has attracted a lot of research in wildlife tourism. The authors refer to other macaque studies but it may be beneficial to include some synthesis of this issue as found in publications by David Newsome or Ronda Green and their colleagues, amongst others in the wildlife tourism discipline.
The work makes for a fascinating study and the authors are to be congratulated as to the clarity of its presentation and the quality of the written work. The style is narrative but this suits the methodology used. I offer a few suggestions to improve the text below.
Line 17: interactions resulted in macaques becoming…
Line 18: intensified
Line 61: livelihoods from the
Line 136: in the tourism
Reviewer 2 Report
This is an exceptionally tight narrative and interesting paper - well done. The Results in particular were well illustrated and descriptive. I do not have anything further for the Discussion or Conclusion, as these are very related to the Results and good extensions of the findings to the context. I do have a few points for further improvement for the authors to consider.
- Possible to set aside your main area of questioning / inquiry into a separate section? It seemed that there was a long and good introduction but then the actual question was missing or perhaps stated in lines 184-185 in a phrasing way. It would be warranted with this level of depth to set aside a paragraph with a header for the research question(s). This would also help make the transition to 3. Materials and Methods easier, as it would preview what the main questions were and then the reader could assess the context and approach for these questions.
- What's the difference between the semi-structured interviews, observations, and informal conversations? More depth is necessary about the differences in the approach and populations for each. Also, the information in the paragraph of lines 224-227 is very vague and encompasses both environmental and social data. This seems out of place, or at least requires more depth to what each part concerns. The final paragraph of the data collection is also unnecessary unless these data were explicitly analyzed. Otherwise, it just seems what would be common in any research context - to know the site.
- Using the process outlined in the Data Analysis, were any codes, themes, or relationships adjusted, added, or deleted? Also, a model of the codes (or at the least, a table) would be very helpful at this point to foreshadow the results. (For example, did it differ from Figure 2?)
- What would future research in this context look like? How would monitoring and managing look like and when would you know that a more stable relationship (or set of relationships across the model) was reached?
